# Biological action at a distance: Correlated pattern formation in adjacent tessellation domains without communication

**John M. Brooke**[1], **Sebastian S. James**[1], **Alejandro Jimenez-Rodriguez**[2], **Stuart P. Wilson**[1]*

**1** Department of Psychology, University of Sheffield, Sheffield, United Kingdom, **2** Department of Computer Science, University of Sheffield, Sheffield, United Kingdom

* S.P.Wilson@sheffield.ac.uk

**Data Availability Statement:** Code for running the simulations reported in this paper is available at https://github.com/ABRG-Models/Tessellations.

**Funding:** This work was supported by a Collaborative Activity Award, Cortical Plasticity

## Abstract

Tessellations emerge in many natural systems, and the constituent domains often contain regular patterns, raising the intriguing possibility that pattern formation within adjacent domains might be correlated by the geometry, without the direct exchange of information between parts comprising either domain. We confirm this paradoxical effect, by simulating pattern formation via reaction-diffusion in domains whose boundary shapes tessellate, and showing that correlations between adjacent patterns are strong compared to controls that self-organize in domains with equivalent sizes but unrelated shapes. The effect holds in systems with linear and non-linear diffusive terms, and for boundary shapes derived from regular and irregular tessellations. Based on the prediction that correlations between adjacent patterns should be bimodally distributed, we develop methods for testing whether a given set of domain boundaries constrained pattern formation within those domains. We then confirm such a prediction by analysing the development of 'subbarrel' patterns, which are thought to emerge via reaction-diffusion, and whose enclosing borders form a Voronoi tessellation on the surface of the rodent somatosensory cortex. In more general terms, this result demonstrates how causal links can be established between the dynamical processes through which biological patterns emerge and the constraints that shape them.

## Author summary

Patterns can form in biological systems as a net effect of dynamical interactions that are excitatory over short distances and inhibitory over larger distances. Patterns that form in this way are known to reflect the shape of the boundary conditions that contain them. But observing that a particular pattern is contained by a boundary is not enough to determine whether or not that boundary was a constraint on pattern formation. Here we develop a novel test for the influence of boundary shape on pattern formation, based on comparing patterns contained by boundaries whose shapes tessellate and thus are geometrically related. Applying this test to patterns of cell density measured in the developing neocortex confirms that cortical column boundaries constrain pattern formation during the first

Within and Across Lifetimes, from the James S McDonnell Foundation (grant 220020516; https://www.jsmf.org/) to SPW. SPW and AJ-R are also supported by the EU Horizon 2020 programme through the FET Flagship Human Brain Project (HBP-SGA3, 945539). The funders played no role in the study design, data collection and analysis, decision to publish, or preparation of the manuscript.

**Competing interests:** The authors have declared that no competing interests exist.

postnatal weeks. In more general terms, our analysis reveals that strong relationships between patterns that form in adjacent biological domains are to be expected based purely on geometrical effects, even if no information is exchanged between those domains during the process of pattern formation. Our analysis provides a means for testing current theories about the fundamental role that constraints play in organising biological systems.

## Introduction

Central to current theories of biological organisation is a distinction between constraint and process. A constraint exerts a causal influence on a dynamical process and is not itself influenced by that process, at the spatial or temporal scale at which those dynamics take place. This definition permits a description of biological function in terms of constraint closure, i.e., the reciprocal interaction of constraints between processes operating at different timescales [1–3] (see also [4, 5]). A step towards falsifying such high-level descriptions of biological organisation is to formulate predictions at the level of specific biological systems, in which those predictions may be tested directly. To this end, our objective here is to operationalize the definition of constraint as causal influence on dynamical process.

The distinction between constraint and process is made explicit in the reaction-diffusion modelling framework [6], which has been successful in accounting for a wide range of biological (and other) phenomena, from the growth of teeth to the spread of tumors and the healing of skin [7, 8]. Reaction-diffusion models describe biological pattern formation in terms of local interactions amongst molecules or cells, which collectively amplify specific modes in an initially random distribution, with those modes determined by the relative size and shape of an enclosing boundary. Hence, the boundary shape is a constraint on the processes of short-range excitation and long-range inhibition from which pattern emerges.

Observing pattern contained by shape therefore suggests that the shape constrained pattern formation. But, alternatively, the enclosing shape may have emerged subsequently to, simultaneously with, or independently of, the formation of the pattern, and it is not obvious how to discriminate between these possibilities. One approach to establishing a causal influence of the boundary on the pattern is by *synthesis*. If the observed shape is imposed as a boundary condition for a reaction-diffusion model, and the evolution of that model gives rise to a similar pattern in simulation, we might infer a causal influence of the shape on the pattern. While compelling and important, such evidence is indirect, as computational modelling is limited to establishing existence proofs for the plausibility of hypotheses, rather than testing them directly. We seek therefore a complementary approach by *analysis* of the pattern, i.e., a direct means of testing between the hypothesis that the shape causally influenced the pattern and the null hypothesis.

To analyse an individual pattern in these terms, one could look for an alignment between the pattern and the boundary shape. For example, incrementing the diffusion constants from an initial choice that amplifies modes of the lowest spatial frequency will, on an elliptical domain, typically produce a sequence of patterns that is first aligned to the longer axis, and subsequently to the shorter axis. Indeed, for a well-defined boundary shape and a simple reaction-diffusion system generating a low spatial-frequency pattern, the alignment of an observed pattern to a hypothetical boundary constraint may be compared with a set of eigenfunctions derived from the linearized equations (i.e., using Mathieu functions for an elliptical domain; [9]). But such methods break down for more complex boundary shapes, for higher-mode

solutions, and for reaction-diffusion dynamics described by increasingly non-linear coupling terms.

In search of a more practical and robust method, the possibility we explore here is to exploit the fact that the shapes of adjacent biological domains are often related to one-another. That is, the processes that determine the shapes of adjacent domain boundaries may themselves be subject to common constraints, or indeed serve as constraints on one-another. Consider the following concrete example. In the plane tangential to the surface of the rodent cortex, the boundary shapes of large cellular aggregates called 'barrels' form a Voronoi tessellation across the primary somatosensory area [10] (see Fig 1). The barrel boundaries are apparent from birth, and from the eighth postnatal day develop 'subbarrel' patterns reflecting variations in thalamocortical innervation density [11] (Fig 1). A reaction-diffusion model, specifically the Keller-Segel formalism with its additional non-linear chemotaxis term, has been used to

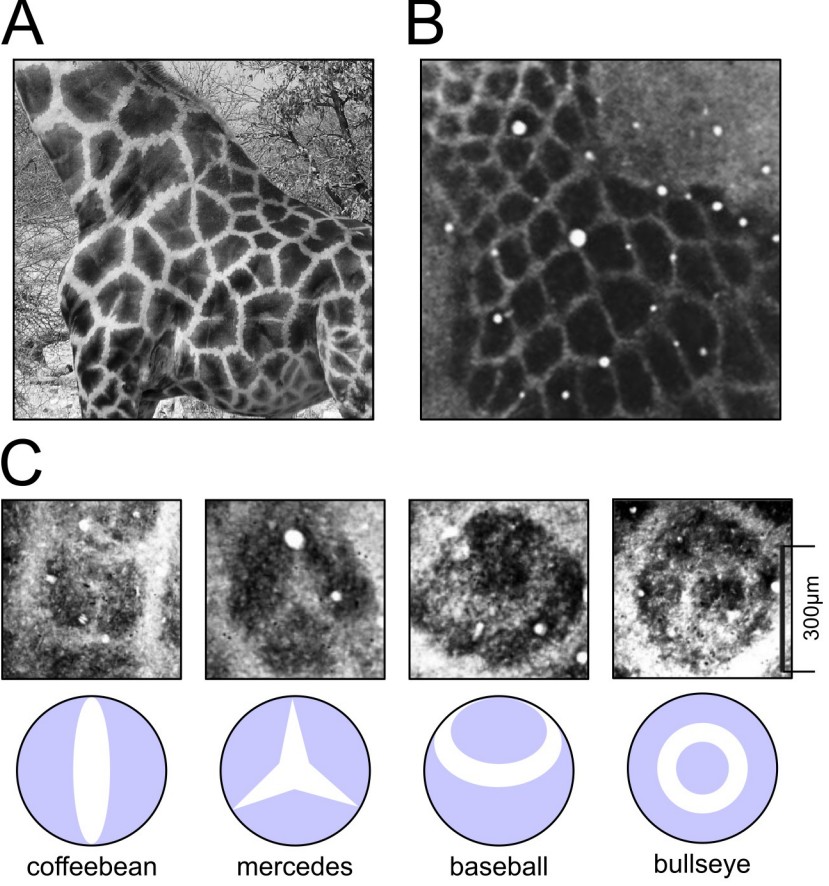

**Fig 1. Tessellating domains and sub-domain structure in biological systems.** At markedly different length scales, the skin of giraffes and the stained neocortices of rodents display similar arrangements of polygonal domains, many of which appear further divided into sub-structures. **A** Image of the skin of a giraffe (*Giraffa camelopardalis reticulata*), credited to O. Berger, and described by Koch & Meinhardt (1994; [56]) as a Voronoi tessellation. The dark panels overlap with a vascular structure that is important for thermoregulation. **B** Image of a tangential section of the cytochrome oxidase stained primary somatosensory cortex of an adult laboratory rat (*Sprague-Dawley*), revealing a pattern of large cortical columns known as 'barrels', which have also been described formally as a Voronoi tessellation. **C** Sub-structures apparent in larger barrel columns have been described in terms of the four categories depicted below, which correspond to the stable patterns generated by a reaction-diffusion model parameterised to amplify modes of increasing spatial frequency. Images in **B** and **C** are from Land & Erickson (2005; [57]) and are shown at a common scale. Photograph in **A** reprinted with permission from Koch A.J. & Meinhardt H., Reviews of Modern Physics, 66, 1481 (1994). Copyright (1994) by the American Physical Society (http://dx.doi.org/10.1103/RevModPhys.66.1481).

successfully recreate subbarrel structure in simulation, as well as to explain an observed relationship between the size of the enclosing barrel boundary and the characteristic mode of the subbarrel pattern ([12]; see also [13]). A synthetic approach has also helped establish that the barrel boundary shapes could emerge to form a Voronoi tessellation based on reaction-diffusion dynamics constrained by the action of orthogonal gene expression gradients on the processes by which thalamocortical axons compete for cortical territory [14]. Hence in this system, the barrel boundary shapes that constrain subbarrel pattern formation via reaction-diffusion are thought also to be related by the common (genetic) constraints under which those barrel boundary shapes emerge.

Within such systems, the geometrical relationship between the shapes of adjacent domain boundaries might be expected to align the patterns that form within those boundaries to a degree that is reflected by the correlation between patterns in either domain. Hence measuring the degree of correlation between adjacent patterns could serve as a proxy for the degree of alignment to the boundary, and thus form the basis of a robust test for the hypothesis that shape constrained pattern formation.

Given chemical, mechanical, and other physical sources of spatial coupling in biological systems, it seems unlikely that pattern formation ever occurs completely independently in proximal and adjacent biological domains. But, in principle, how much of a relationship between patterns that form within adjacent domains might we expect to observe under the assumption that no communication occurs across domain boundaries?

On face value, this question might seem misguided. If pattern formation amongst cells within a particular domain occurs without the direct exchange of information with cells of an adjacent domain, then on what basis should we expect to measure any relationship at all between the patterns that form within adjacent domains? As we will show, strong correlations between patterns that self-organize independently in adjacent domains are in fact to be expected, if the shapes of those domains are geometrically related. Specifically, correlations are to be expected if the boundaries of adjacent domains abut, such that the domain shapes constitute a tessellation. Simulation experiments and analyses reported herein are designed to establish how relationships between domains on the basis of their shapes and common boundary lengths contribute to this somewhat paradoxical effect.

## Results

The key insight developed here is that patterns that self-organize independently in adjacent domains of a tessellation should nevertheless be correlated. Hence, by analysing the correlations between patterns measured in adjacent domains we can directly test the hypothesis that those observed patterns self-organized under constraints imposed by the borders within which they are observed to be enclosed. We will demonstrate the robustness of the (predicted) correlation effect by examining numerical solutions to reaction-diffusion equations that have been evaluated in domains that tessellate under a range of different geometrical constraints. We will begin with an instructive toy example that will reveal the correlation effect most clearly. We will then show by analysis that the effect holds in a specific biological case (subbarrel patterning).

The patterns on which we will base our analyses can be generated on the two-dimensional plane $\mathbf{x}$ by solving reaction-diffusion equations of the form

$$
\begin{aligned}
\frac{\partial n(\mathbf{x}, t)}{\partial t} &= 1 - n(\mathbf{x}, t) + D_n \nabla^2 n(\mathbf{x}, t) - \chi \nabla.(n(\mathbf{x}, t) \nabla c(\mathbf{x}, t)) \\
\frac{\partial c(\mathbf{x}, t)}{\partial t} &= f(n(\mathbf{x}, t)) - c(\mathbf{x}, t) + D_c \nabla^2 c(\mathbf{x}, t)
\end{aligned}
\tag{1}
$$

where $n$ and $c$ are two interacting species, $D_n$ and $D_c$ are diffusion constants, and the 'chemotaxis' term $\chi$ specifies the strength of the interaction between the two species. Following [12] we will use $f(n) = \gamma \frac{n^2}{(1+n^2)}$ with $\gamma = 5$, and set $D_n \leq \chi$.

## Bimodal pattern correlations amongst adjacent domains signal boundary constraints

Consider a reaction-diffusion system (e.g., Eq 1) constrained by a boundary in the shape of an equilateral triangle (Fig 2A). Solved for a choice of diffusion constants that yield patterns with the lowest spatial-frequencies, this system will generate one of two basic kinds of pattern (e.g., in the concentration of $n$ and/or $c$). In the first, one of the extreme values of the reaction, positive or negative, will collect in one of the three corners of the triangle and values at the other extreme will be spread out across the opposite edge. Along that edge the values are essentially constant, and along the other two edges the values vary from extreme high to extreme low. In the second kind of pattern, values at the two extremes will collect in two corners and values around zero will collect in the third. Along one edge the values vary from extreme high to extreme low and along the other two they vary from zero to either extreme. Values sampled

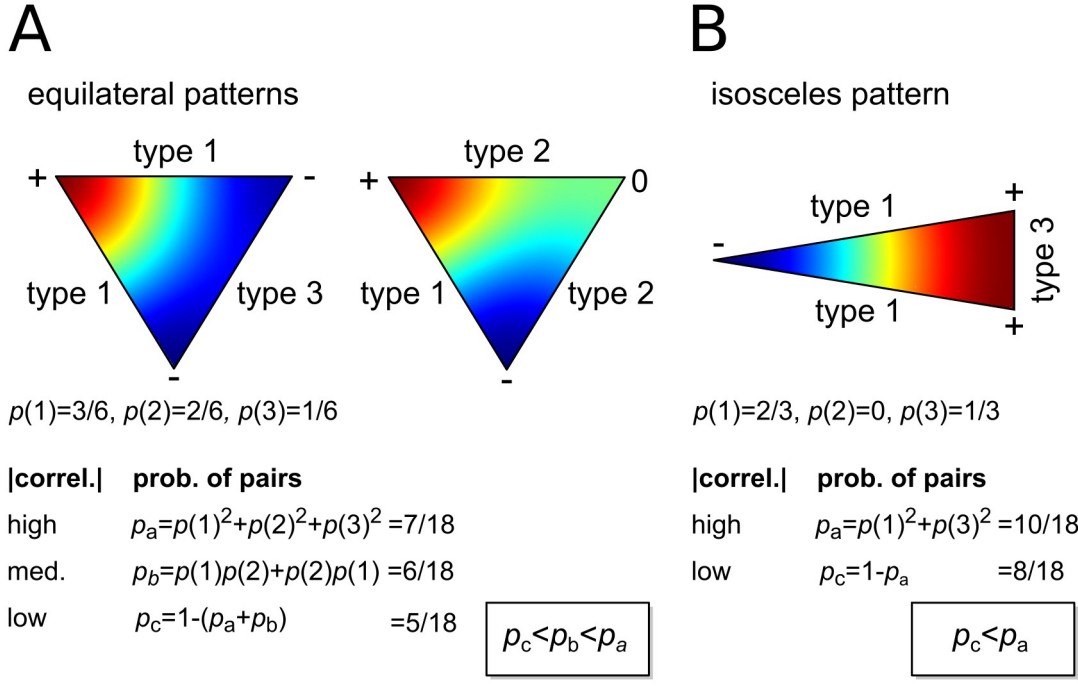

$p(1) = 3/6,\ p(2) = 2/6,\ p(3) = 1/6$

$p(1) = 2/3,\ p(2) = 0,\ p(3) = 1/3$

| |correl.| | prob. of pairs | |
|---|---|---|
| high | $p_a = p(1)^2 + p(2)^2 + p(3)^2$ | $= 7/18$ |
| med. | $p_b = p(1)p(2) + p(2)p(1)$ | $= 6/18$ |
| low | $p_c = 1 - (p_a + p_b)$ | $= 5/18$ |

$$p_c < p_b < p_a$$

| |correl.| | prob. of pairs | |
|---|---|---|
| high | $p_a = p(1)^2 + p(3)^2$ | $= 10/18$ |
| low | $p_c = 1 - p_a$ | $= 8/18$ |

$$p_c < p_a$$

**Fig 2. The distribution of pattern correlations along common edges of tessellating triangles should be bimodal.** Colour images show typical patterns generated by a reaction-diffusion model with a large diffusivity term, using a colour map in which red and blue mark extreme high and low concentration values, and green marks zero concentration. **A** Solved within the boundary of an equilateral triangle, two basic patterns emerge, with extreme concentrations in one corner and along the opposite edge (left) or at two corners (right). Along the edges, three pattern types are apparent. Type 1 varies between the two extremes, type 2 varies between one extreme and zero, and type 3 does not vary. The probability of type $i$ is given below as $p(i)$. The table gives the probability that the absolute correlation between patterns sampled along two randomly chosen edges will be high ($p_a$), medium ($p_b$), or low ($p_c$). As $p_c < p_b < p_a$ the distribution of correlations should be bimodal. **B** Patterns that emerge within the boundary of an isosceles triangle will be of type 1 or 3 only, changing the distribution of correlations across random edge pairs while retaining an overall bimodal distribution ($p_c < p_a$). However, if pairs of edges are restricted to those which may be adjacent in a tessellation then only pairs of type 1 and pairs of type 3 are possible, and $p_a = 1$. Hence, in more general terms, the distribution of correlations between patterns measured along the edges of *adjacent tessellation* domains should be even more strongly bimodal.

along the edges will vary between the two extremes in $\frac{3}{6}$ of the edge types (type 1), they will vary from one extreme to zero for $\frac{2}{6}$ of the edge types (type 2), and they will not vary along the edge for $\frac{1}{6}$ of the edge types (type 3). Assuming (for simplicity) that the two kinds of pattern occur equally often, and that pairs of edges are drawn at random from a large enough sample, two of the same edge type will be drawn with a probability that tends toward $p_a \rightarrow \frac{1}{2^2} + \frac{1}{3^2} + \frac{1}{6^2} = \frac{7}{18}$ ('tends to' denoted by $\rightarrow$). A type 1 and type 2 edge will be paired with a probability of $p_b \rightarrow 2\frac{1}{6} = \frac{6}{18}$. And a type 3 edge will be paired with a type 1 or 2 for the remaining $p_c \rightarrow \frac{5}{18}$. Now consider that along the edge, the magnitude of the correlation between the values sampled will be high for $p_a$ pairs, low for $p_c$ pairs, and intermediate for $p_b$ pairs. Given that $p_c < p_b < p_a$, that the magnitude of each correlation level increases with its probability of occurring, and that correlations and anti-correlations at each level are equiprobable given the symmetries within each kind of pattern, the distribution of correlations should be (overall) bimodal. Note that we describe the distribution as *overall* bimodal because smaller secondary peaks are expected to emerge around each distinct correlation level.

Consider next what happens when we substitute equilateral triangles with isosceles triangles (Fig 2B). Reducing the number of axes of symmetry from three to one further constrains the kinds of patterns that are possible, causing (low spatial-frequency) solutions of the reaction-diffusion system to align with the perpendicular bisector of the base, and reducing the pattern along the edges to two types only. For example, if the base is the shorter side then $\frac{2}{3}$ of the edges will be of type 1 and $\frac{1}{3}$ will be of type 3. Pairs of the same type constitute $p_a \rightarrow \frac{10}{18}$ and pairs of different types constitute $p_c \rightarrow \frac{8}{18}$, so again $p_c < p_a$ and the distribution should again be (overall) bimodal. We note two important differences between the equilateral and isosceles cases. First, as pattern formation is more constrained by the isosceles boundary shape, and so the number of different kinds of patterns that are possible is reduced, the proportion of extreme correlations (and anti-correlations) has increased, from $p_a \rightarrow \frac{6}{18} = 0.333$ to $p_a \rightarrow \frac{10}{18} = 0.556$. Second, the number of secondary peaks in the distribution of correlations has reduced to just two, around the positive and negative correlations corresponding to $p_c$.

Because the pattern in each triangle is independent, any equilateral triangle in a tessellation can be substituted or rotated so that a given edge is adjacent to any other. Hence we expect to sample from the same distribution of correlations whether we choose pairs at random, or limit our choices to those edges that are adjacent. This is not the case for the isosceles triangles, which only tessellate by arranging neighbours base-to-base or with the bases' perpendicular bisectors antiparallel. A base cannot be adjacent to a non-base, and hence the distribution of correlations obtained from sampling adjacent pairs will lose its secondary peaks to display only the highest correlations and anti-correlations. So correlations sampled from adjacent rather than randomly selected edge pairs should be even more strongly bimodal.

Further, imagine randomly displacing each vertex of the tessellation of equilateral triangles in order to construct an irregular tessellation of scalene triangles (Fig 3C). As each vertex is common to three triangles, each displacement changes the constraints on pattern formation in three triangles, from an initial minimally constraining configuration, and as such, increases the overall bimodality of the distribution of correlations. The irregular tessellation permits no substitution of domains, and hence, as in the isosceles case, we expect the overall bimodality of the distribution of correlations to be greater when comparing patterns amongst adjacent edges compared to randomly chosen edges.

An overall bimodal distribution of correlations amongst values sampled along pairs of edges from adjacent domains is therefore to be expected for domains that tessellate either regularly or irregularly. This property indicates that the domain boundaries constrained pattern formation. As a final thought experiment, consider that a jigsaw puzzle, i.e., an image into

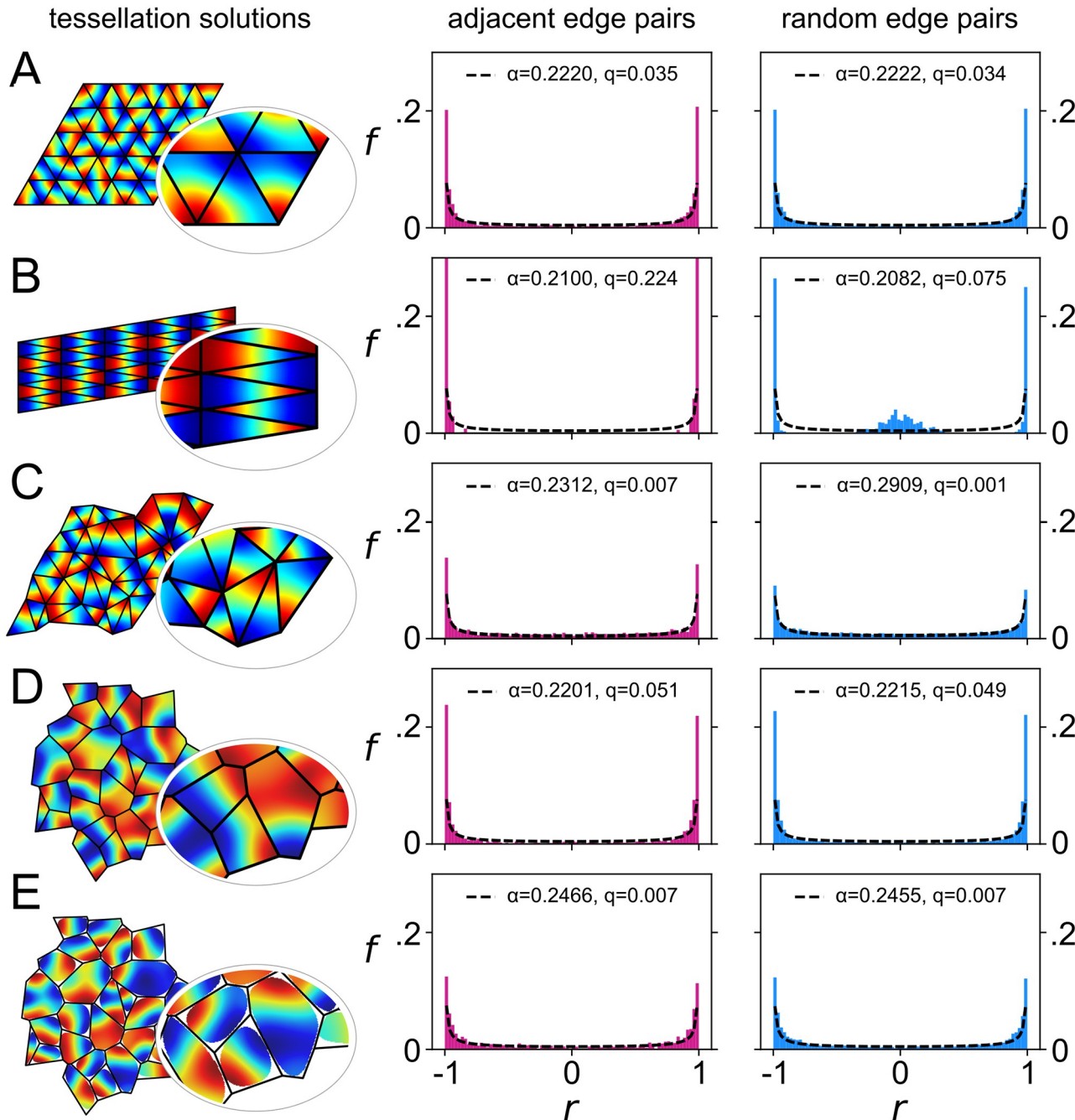

**Fig 3. Correlated pattern formation in adjacent tessellation domains without communication.** A system of reaction-diffusion equations (Eq 1; $D_n = \chi = 36$) was solved using boundary shapes that tessellate in different ways (left column), with blue and red corresponding to extreme positive and negative values, and black lines delineating the domains. Values were sampled along the individual vertices of each domain and samples were correlated between edges of *different* domains, either amongst pairs of edges that are adjacent in the tessellation (center column) or randomly selected (right column). Histograms show the distributions ($f$) of correlation coefficients ($r$) obtained in either case, which were fit by the beta-distribution (dotted line) parameterized by $\alpha$ (see text for details; $q$ is the sum of squared differences between the data and the fit). Rows **A**-**E** show data obtained from tessellations comprising domains with different shapes: **A** equilateral triangles; **B** isosceles triangles; **C** scalene triangles; **D** a Voronoi tessellation; **E** a Voronoi tessellation with rounded vertices. Peaks at ±1 in the histograms indicate that while pattern formation occurs entirely independently within each domain, patterns may become correlated between (adjacent) domains due to common constraints that derive from the fact that their boundary shapes tessellate.

which borders are subsequently cut, will display perfectly strong positive correlations across adjacent edges and no anti-correlations. But our considerations thus far suggest that strong correlations and anti-correlations should be equally likely when the tessellation boundaries constrain subsequent pattern formation. Thus it is really the presence of strong anti-correlations in the distribution that evidences a causal influence of domain shape on pattern formation.

## Correlated pattern formation in adjacent domains of naturalistic tessellations

To test our reasoning we solved the reaction-diffusion system defined by Eq 1 numerically, using a dense hexagonal lattice of grid points, in domains with boundary shapes that were either equilateral triangles (Fig 3A), isosceles triangles (Fig 3B), or scalene triangles (Fig 3C), which in each case could be fit together to form a tessellation. Pattern formation was simulated independently in each triangular domain using no-flux boundary conditions ($D_n = \chi = 36$, $D_c = 0.3D_n$), and settled values of $n$ were sampled along the edges of each boundary, one hexagonal grid point in from the edge. Pearson correlation coefficients were then calculated from samples taken either along the edges of domains that were adjacent on the tessellation, or from randomly chosen edge pairs. The distributions of correlation coefficients obtained from random and adjacent edges were compared in each case using a Kolmogorov-Smirnov test (see [15]). For equilateral triangles this analysis revealed no significant difference, and for isosceles triangles ($p < 0.001$) and scalene triangles ($p < 0.001$) the difference was highly significant, as anticipated.

Considering pattern formation on tessellations of triangles is instructive, but to what extent do the considerations developed here apply to the kinds of tessellation observed in natural systems?

Examples of Voronoi tessellations are commonly found in the natural world [16–18], including the packing of epithelial cells, the patterning of giraffe skins, and modular structures in the functional organization of the neocortex. The domains of a Voronoi tessellation enclose all points that are closer to a given 'seed point' than any other. As such, the polygonal structure of the tessellation is completely specified by a collection of seed points, with points along the polygonal boundaries equidistant to two seed points and points at the vertices equidistant from three. To test whether the predicted bimodal correlation is also to be expected in these naturally occurring tessellation structures, we generated random Voronoi tessellations from randomly chosen seed point coordinates, and solved the reaction-diffusion system (independently) within each domain. As shown in Fig 3D, the distribution of correlations sampled from along adjacent edges is again clearly overall bimodal. Hence, the effect is not specific to the case of triangles, and is to be expected for irregular tessellations of polygons that have a range of different numbers and arrangements of vertices.

The domains that comprise naturally occurring tessellations are often "Dirichletiform" ([10], p. 350), but may not be strictly polygonal, with rounded corners rather than definite angles at the vertices [19]. And it is known that patterns formed by reaction-diffusion systems tend to be strongly influenced by the presence of definite angular intersections at the vertices [20]. Therefore, to establish whether bimodality is also predicted for such natural structures, we re-constructed the random Voronoi tessellation and rounded the corners of the domains by joining the midpoints of each edge with quadratic Bezier curves whose first derivatives fit continuously at the midpoint. We then reconstructed the edges that corresponded to those of the original polygon by recording the points where the radial segments joining the centroid of the original polygon to its vertices cut the new shape. The reaction-diffusion system was solved

again on the resulting domains, and (rounded) edges in the same locations as for the analysis of the original (polygonal) tessellation were correlated for a direct comparison. As shown in Fig 3E, the distribution of correlations along adjacent edges is again predicted to be bimodal. Hence, the effect is not specific to polygonal domains and indeed is to be expected in this more general case.

Kolmogorov-Smirnov tests revealed that the distributions of correlations sampled from adjacent versus random edge pairs were not significantly different when Voronoi domains were strictly polygonal or when the domain edges were rounded.

## Measuring the effect of boundary constraints on pattern formation and alignment

In order to measure the degree to which the boundaries constrain pattern formation, we consider the known result (see e.g., [21] Ch. 4) that the scalar products of vectors that are uniformly randomly distributed on a unit hypersphere of dimension $D - 1$ (i.e., embedded in a space of dimension $D$) follow the beta distribution, $\mu_{B}(u)$, on $u \in (0, 1)$,

$$\mu_{B}(u) = \frac{u^{\alpha-1}(1 - u)^{\beta-1}}{B(\alpha, \beta)}, \tag{2}$$

with $\alpha = \beta = (D - 1)/2$, and $B(\alpha, \beta)$ the standard beta function [9]. It can be seen that the beta distribution diverges at $u = 0$ and at $u = 1$ if $D = 2$, i.e., when the vectors are uniformly distributed on the unit circle, but that it conforms to a uniform distribution for a sphere in 3 dimensions ($D = 3$). Note that these dimensions pertain to the abstract vector space of all normalised edge vectors, and hence the dimension can in principle be as large as the numerical discretization that the tessellation permits. However, the coherence of the vectors derived from the smoothness of the solutions of Eq 1 ensures that they lie in subspaces of much lower dimension. We measured correlations using the Pearson correlation coefficient, which is equivalent to calculating the dot product of two unit vectors, and thus we can use simple algebra to map from the domain $[-1, +1]$ to $[0, 1]$. If the edge vectors are not 'pinned' to the tessellation we expect them to be able to 'slip' relative to each other so they become uniformly distributed on a circle, and consequently $\alpha \to 0.5$ (see Methods for a proof). Estimates of the corresponding symmetric ($\alpha = \beta$) beta distribution fits are shown with the histograms in Fig 3, where $\alpha < 0.5$, from which we deduce that they are not uniformly distributed, exactly as expected if the influence of the tessellation on pattern formation were to preferentially select certain mutual orientations along adjacent edges. For completeness we note that replacing the coherent fields generated by reaction-diffusion with fields that have random values, and thus no spatial pattern, instead gives a distribution that approaches a normal distribution ($\alpha \to \infty$).

## Correlations are not bimodally distributed if borders are imposed after pattern formation

So far we have considered only the lowest mode solutions produced by a reaction-diffusion system. To explore whether the results should hold for the more complicated patterns that may be produced by more complex pattern-forming systems, we conducted a sweep of the parameter space, varying the diffusion parameter $D_n$ and the parameter in Eq 1 that weights the contribution of the non-linear coupling term, $\chi$, while keeping $D_c = 0.3D_n$ throughout (Fig 4). For each parameter combination we solved the reaction-diffusion equations on ensembles of domains from randomly seeded Voronoi tessellations.

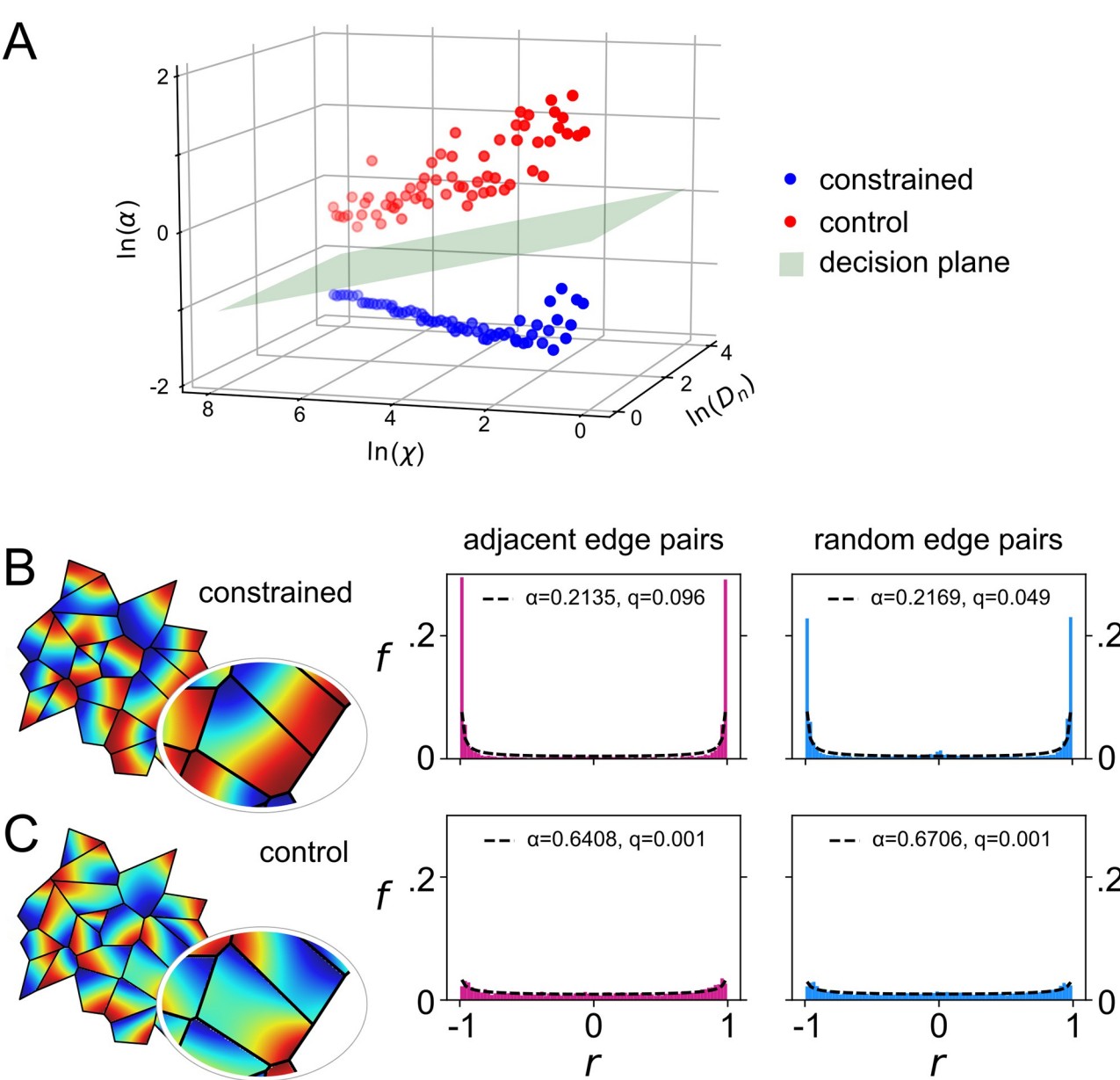

**Fig 4. Analysis of control patterns formed without shaped boundary constraints registers only very weak correlations.** Combinations of eight values of the diffusion constant $D_n$ and eight values of the constant $\chi$ that weights the non-linear coupling term were evaluated on domains of a Voronoi tessellation generated from random seed points. The remaining free parameter, $D_c$ was set to $0.3D_n$, and increments in $\chi$ were expressed as proportions of $D_n$ to cover a large parameter space. In the 'constrained' condition, the shapes of the domain boundaries were a constraint on pattern formation. In the 'control' condition, solutions were obtained in circular domains, and the tessellation boundaries were imposed only after pattern formation, to allow a corresponding set of correlations to be measured for comparison. **A** Values of $\alpha$ were obtained in each condition and for each parameter combination by fitting the resulting distribution of adjacent-edge correlations. Only weak bimodality (high $\alpha$) was observed in the control condition. Following a log transformation to each axis, $\alpha$ values were clearly linearly separable, as confirmed by the success of a perceptron in discriminating the two conditions (perceptron decision boundary shown in green). Example solutions in the constrained and control conditions are shown in **B** and **C**, respectively, for the combination of parameters ($D_n = \chi = 36$) that yielded the lowest $\alpha$ values in the control condition ($\alpha = 0.64$).

It was conceivable to us that a degree of correlation (and anti-correlation) may be expected due to chance for patterns of low spatial frequency, even without the boundary shape constraining pattern formation. So we also solved each system of equations on an ensemble of *circles*, centred at locations derived from the original Voronoi tessellation seed points, but

subjected to additional random displacement by vectors whose radii and polar angles were chosen from a uniform distribution, normalised so that the new centres remained inside the original polygons. The size of each circle was chosen so that it minimally overlapped with the corresponding polygon from the original tessellation. We then overlaid the original tessellation onto the ensemble of circles, extracted the field values along the overlaid edges, and obtained the distribution of correlations for each case as previously described. The purpose of this procedure was to remove any possible influence of domain shape while ensuring that the data subjected to analysis were sampled from regions that tessellated precisely.

First we consider the case where $D_n = \chi = 36$ for the constrained condition (Fig 4B) and the control condition (Fig 4C). Visual inspection of the alignment between the control patterns appeared similar to that between patterns formed under the constraints of the polygonal boundaries. However, histograms of the distribution of the correlations showed that they were quantitatively different. The value of $\alpha$ obtained in the boundary-constrained condition was well below the threshold value of 0.5, as expected. By contrast, the histogram obtained in the control condition (Fig 4C) yields $0.5 < \alpha < 1.0$. Since the patterns that formed in this condition were not constrained by the tessellation, the increase in the degrees of freedom of their relative orientations produced a distribution that lost most of the bimodality and which thus approaches the uniform distribution.

Next we consider how the two distributions of adjacent-edge correlations vary across the full range of parameters. Following log transformations of $D_n$, $\chi$, and $\alpha$, data obtained from simulations run in the control and boundary-constrained conditions were linearly separable across the full range of parameter values tested (Fig 4A). To confirm this we trained a perceptron to discriminate between control (target response $y = 0$) and boundary-constrained ($y = 1$) data. Training vectors $\mathbf{x} = [\ln(D_n), \ln(\chi), \ln(\alpha), 1]$ were presented in a random sequence and the perceptron weights $w_i$ were updated following each presentation using the delta rule: $\Delta w_i = \epsilon(y - \sigma(u))\sigma'(u)x_i$, where $\sigma(u) = (1 + e^{-u})^{-1}$, $\sigma'(u) = \sigma(u)(1 - \sigma(u))$, and $\epsilon = 0.05$. The resulting weights ($\mathbf{w} = [0.01, -0.46, -3.26, 0.21]$) define a decision boundary, where $u = \mathbf{x} \cdot \mathbf{w} = 0$, shown as a plane in Fig 4A that clearly separates the data obtained from the two conditions.

In Fig 5, the contour line corresponding to the analytical threshold ($\alpha = 0.5$) runs approximately diagonally across the region, and is effective at distinguishing the influence of the tessellation over more than two thirds of this large parameter space. Example fields and the associated estimates of $\alpha$ are shown for the four extreme corners of the parameter space in Fig 5B. Two are within the region where the threshold can detect the effect of the tessellation on the solutions. Towards the top, where $D_n$ is low, the fields become very concentrated and the nonlinear gradients in the region are so strong that the effects of the boundaries are not transmitted to the interior. However when $D_n$ is larger, parameters that yield complex fields that reflect the amplification of several modes clearly support the hypothesis that the tessellation boundaries constrained pattern formation.

It is possible that amongst the domains of a biological tessellation the control parameters for self-organisation may show some variation. To determine the robustness of the reported effects we therefore re-ran simulations with parameters in the mid-range of the space that was tiled by our initial parameter sweep ($D_n = \chi = 6.0$, $D_c = 0.3D_n$) and then randomly perturbed these values in each domain by up to 10%. Compared to the unperturbed case, distributions of (adjacent) correlations were not statistically different (Kolmogorov-Smirnov test, $p = 0.997$). When the control parameters were perturbed by up to 50% in each domain, the correlations appeared to diverge a little, but not enough to reject the null hypothesis that they were drawn from the same distribution ($p = 0.31$). Moreover, the effects shown in Fig 5 are not sensitive to the particular choice of pattern-forming system, as confirmed via a sweep through the relevant

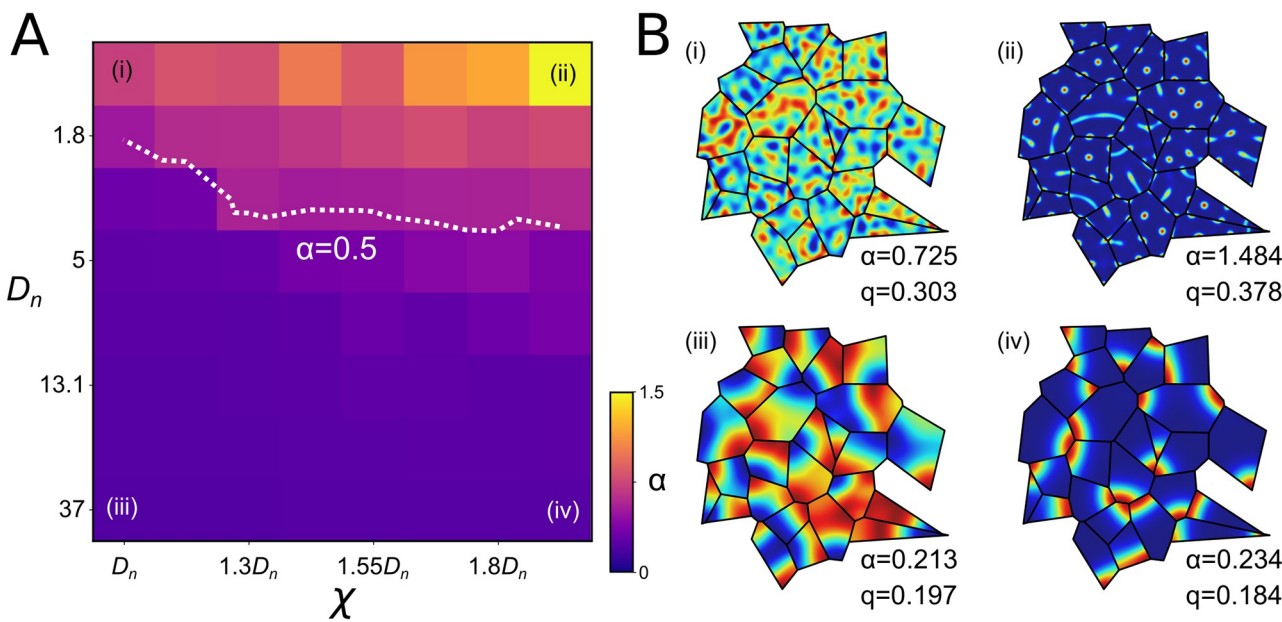

**Fig 5. Correlated pattern formation in tessellated domains is predicted to emerge robustly across a wide range of pattern-forming systems.** Correlations between self-organized patterns in adjacent domains of randomly seeded Voronoi tessellations were measured across a wide range of parameters. Panel **A** shows values of $\alpha$ estimated from the distribution of 1000 pairwise correlation coefficients obtained from each of sixty four combinations of parameter values (as in Fig 4A; 'constrained'). The overlaid contour corresponds to the threshold, $\alpha = 0.5$, at or below which the hypothesis that the domain boundaries constrained pattern formation is very strongly supported. Based on this threshold, patterns are expected to be correlated by the tessellation boundary constraints across a large portion of the parameter space. Panel **B** shows example patterns for four extreme cases.

parameters of an alternative system that does not include a non-linear diffusion coupling term of the type that is parameterised by $\chi$ in Eq 1 (see S1 Fig; [22, 23]).

## Emergence of bimodal correlations confirms that column boundaries constrain thalamocortical patterning in the developing barrel cortex

The emergence of subbarrel patterns of thalamocortical innervation density in the rodent somatosensory cortex has been successfully modelled using the Keller-Segel reaction-diffusion system [13], with the borders of individual barrels imposed as a boundary constraint on pattern formation [12] (see Introduction and Fig 1). The barrel borders form a Voronoi tessellation, though the edges are typically a little rounded [10]. The barrel structure is present from birth and the subbarrel patterns are first apparent at around postnatal day 8, and become clearly defined by around postnatal day 10, in stains for seretonin transporter and other markers for synaptic activity [11]. If subbarrel patterns emerge via reaction-diffusion dynamics under the constraints of the barrel boundaries, our analysis predicts that we should see a bimodal distribution of correlations along the common edges of adjacent barrels.

To test this hypothesis, we analysed three images of seretonin transporter expression reported by Louderback and colleagues ([11]; their Figure 4). The results of the analysis are shown in Fig 6. We developed a simple computer program to sample the average image pixel intensity in rectangular bins pointing outward-normal to the two parallel sides of a user defined rectangle. Using this tool we defined rectangles to coincide with line segments corresponding to the septal regions that separate the barrels, then for each segment sampled from fifty bins along the outer edge of two adjacent barrel regions, to a depth of twenty pixels ($\sim 85$ $\mu$m) into each of the barrels. Care was taken to ensure that the length of each line segment was

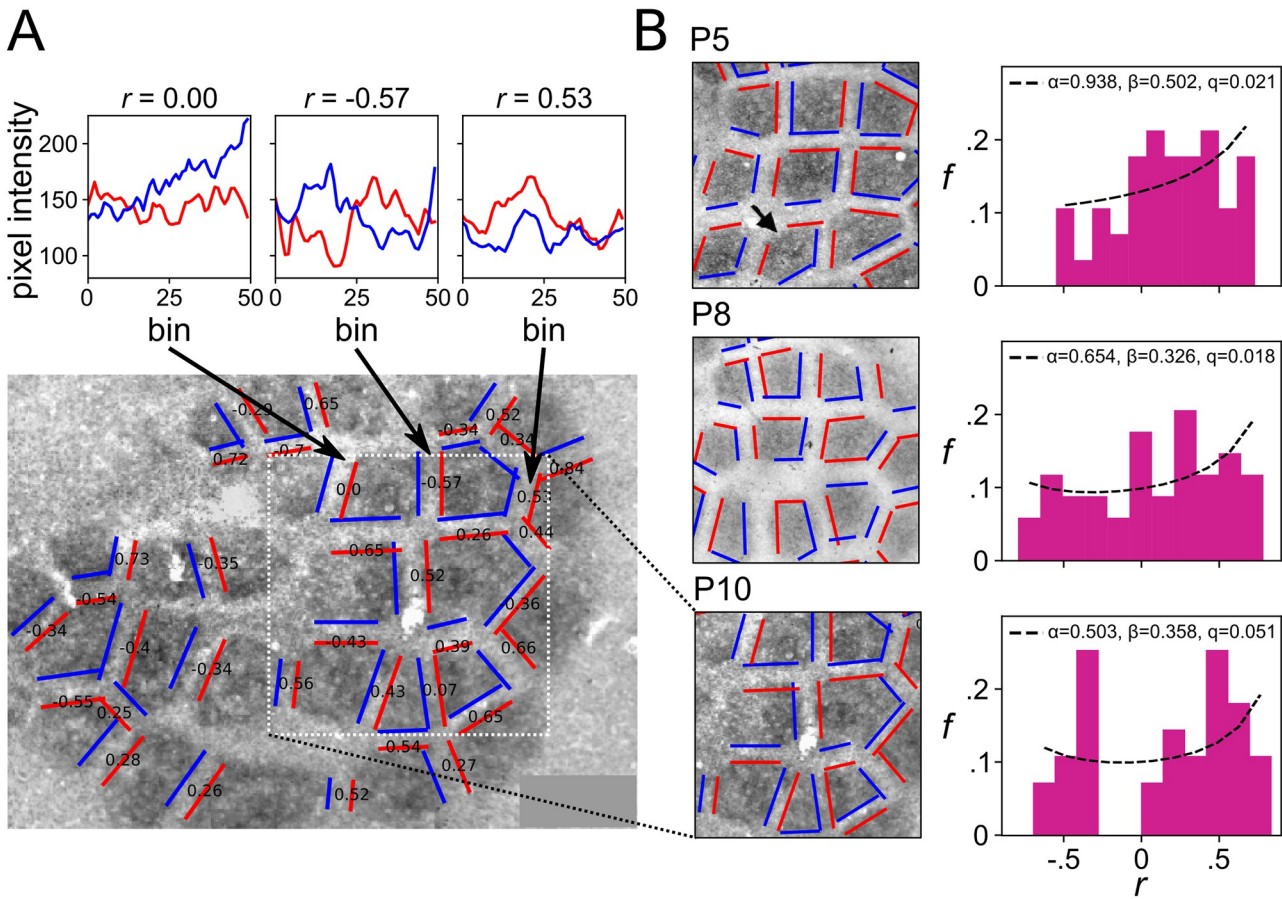

**Fig 6. Emergence of correlated patterns in adjacent domains of the developing neocortex.** We analysed images of immunohistochemical stains for serotonin transporter (5-HTT) expression on the surface of the rat barrel cortex, obtained at postnatal days 5 (P5), 8 (P8), and 10 (P10). This stain reveals the shapes of the barrel columns, each corresponding to a whisker on the animal's snout, as large dark polygonal patches forming a Voronoi tessellation. From P8, sub-barrel structures become apparent and by P10 they clearly identify several regions of high synaptic density within many barrels. Panel **A** shows the details of the analysis method for the P10 image. Overlaid pairs of parallel red and blue lines show the extents along which image intensity was sampled for each pairwise comparison. Each line marks a vertex of the barrel boundary, and samples were constructed by averaging the grayscale intensity of pixels in one of 50 regularly spaced rectangular bins extending a short distance in from the line towards the corresponding barrel center. The correlation coefficient for each pair of samples is shown in black text, and the plots above show sampled data used for three example pairwise comparisons. Distributions of correlation coefficients obtained from pairs of edges from adjacent barrels are shown for each postnatal day in **B**, showing a clear progression from a unimodal shape at P5 to a bimodal shape at P10, and supporting the hypothesis that pattern formation within the barrels occurs postnatally and is constrained by the barrel boundary shapes. Original images from [11].

as long as possible (to include as much of the border as possible), and that the width of each rectangle was as short as possible (to sample as close to the border of the barrels as possible), while not sampling from the septal region itself (to avoid introducing light/dark transitions into the sample that could cause spurious positive correlations). A small number of adjacent edges were excluded as their edges were not clearly parallel, but overall good coverage of the boundaries was achieved.

Examples of the variation in pixel intensity along sampling bins spanning parallel line segments of adjacent barrels are shown at the top of Fig 6A, revealing clear correlations and anticorrelations at postnatal day 10. In real data like this, it is conceivable that the technique could pick up spurious correlations, for example if image artefacts appeared in the sample from both edges of a pair, but we note that visible artefacts (e.g., circular bubbles of light or dark related to the underlying vasculature) very rarely spanned the width of the septa and when they did

were very rarely located in or around the septa. Moreover, as noted above, anticorrelations are not to be expected by chance.

Images obtained from rats at postnatal days 5, 8, and 10 were analysed. At postnatal day 5, prior to when subbarrel patterns are reported to emerge, we found the distribution of adjacent-pair correlations in seretonin expression to be unimodal, about a mean value of $0.18 \pm 0.34$. At postnatal day 8, when subbarrel patterns are reported to become apparent, two distinct peaks at a correlation of approximately $\pm 0.5$ were also apparent. At postnatal day 10, when subbarrel patterns are reported to be well defined, and were clearly visible in the image of seretonin expression, the distribution was clearly bimodal, with essentially all pairs showing non-zero correlations. Only the P10 distribution failed a test of unimodality (Hartigan's dip test; $p = 0.01$). Fitting the distribution of correlations using Eq 2 yielded an estimate of $\alpha = 0.94$ and $\beta = 0.5$ at postnatal day 5 (P5), an estimate of $\alpha = 0.65$ and $\beta = 0.33$ at P8, and $\alpha = 0.50$ and $\beta = 0.36$ at P10. While the theory predicts that the distributions should be symmetric, i.e., $\alpha = \beta$, the larger estimates for $\alpha$ reflect a general shift in each distribution to the right due to additional sources of positive correlation to be expected when extracting a fairly small sample from image data, as previously noted. As $\alpha$ is the parameter more sensitive to the presence of anticorrelations, we interpret its decrease, by P10, to a value that strongly supports the hypothesis that the domain boundaries constrained pattern formation, as strong evidence that subbarrel patterns emerge postnatally under constraints imposed by the barrel boundary shapes.

Thus our analysis supports the model of subbarrel pattern formation as a product of reaction-diffusion dynamics constrained by the barrel boundary shapes [12]. Moreover, this result demonstrates how the definition of constraint as a causal influence on biological process can practically be operationalized in terms of the distribution of adjacent pairwise pattern correlations, for reaction-diffusion systems on tessellated domains.

## Discussion

We have shown that because the shape of a domain boundary aligns pattern formation via reaction-diffusion, pattern formation within adjacent domains of a tessellation gives rise to an alignment between those patterns that can be measured as a strong (anti-)correlation between cells located on either side of a common boundary. Our simulation results demonstrate that the alignment of patterns in adjacent domains is predicted to be robust, with alignment occurring over a wide range of length scales, as set by the diffusion constants, and in reaction-diffusion systems both with and without non-linear coupling of the dynamic variables (Fig 5 and S1 Fig). They also demonstrate that while rounding the vertices of the domains reduces the effect, it does not destroy it, and hence alignment is likely also to occur in biological domains where the boundary shapes may be less strictly polygonal (Fig 3E). Our results show that the effect is not to be expected in tessellated domains whose boundaries did not constrain pattern formation (Fig 4B). Hence they establish how bimodality in the distribution of correlations measured across adjacent edges of a tessellation (and specifically the presence of anti-correlations) can be used to test the hypothesis that the domain boundaries constrained pattern formation via reaction-diffusion. This hypothesis was confirmed by the analysis of patterns of cell density that are thought to be formed via reaction-diffusion dynamics in the rodent somatosensory cortex, as a specific example system (Fig 6).

The alignment effect is paradoxical, and an interesting biological example of action at a distance, because the process of pattern formation within a given domain occurs entirely independently of pattern formation in any other, and thus it involves no communication between cells that are located in different domains. Yet the effect is quite understandable, in geometric terms, when we consider that the boundary conditions of a given domain implicitly contain

information about the boundary conditions of other domains, in the knowledge (or under the assumption) that those domains tessellate, and hence are related by a common underlying causal structure; e.g., by the collection of seed points from which a Voronoi tessellation originates.

Ours is not the first demonstration that biological variables can become synchronised in spatially separate populations that do not communicate directly. For example, it is well known that the similarity in weather patterns between two locations tends to decrease with the distance between them. And as such, population dynamics in two separate groups of conspecifics that do not interact will tend to be correlated if their habitats are nearby, due to the effects of a common weather pattern as a mediating 'third' variable. This is an example of the "Moran effect" [24–27]. The effect we have described is distinct in two ways. First, it arises from comparisons between the structures of patterns that vary in space and time (i.e., in systems described by partial differential equations), whereas the Moran effect describes temporal fluctuations only and is thus typically modelled using systems of ordinary differential equations. Secondly, the correlations studied here originate from the *constant* effect of the boundary conditions on pattern formation, rather than by the common influence of any time-varying quantity. We expect that both effects may yet be understood as instances of a more general class of phenomena by which spatial relationships between environmental variables (i.e., in terms of their proximity or boundary shapes) induce correlations between otherwise decoupled populations. To this end, future investigations may extend the analysis developed here to the study of systems whose solutions oscillate in both space and time.

Indeed, the potential importance of the effect established here for understanding biological organization comes into focus when we consider how such causal structures might interact at different timescales [28, 29]. Specifically, how might the alignment of patterns by their boundary constraints in turn constrain the slower processes that are involved in maintaining those boundary constraints? We can think of two broad answers, relating to the affordances of pattern alignment for material transport, and for information processing, though there may be several more.

In terms of material transport, if the pattern of concentration produced by a reaction-diffusion system corresponds to the density of cells or other physical obstacles, as it does in the example of neocortical patterning, then correlations along a common boundary edge create, in the regions of low concentration, channels through which other materials may flow. Uncorrelated patterns, such as those generated by our control simulations (Fig 4B), are discontinuous at all borders and here create bottlenecks that restrict the flow of small materials and stop the flow of larger materials. In these terms, the anti-correlations that come with pattern alignment are of course bottle tops, permitting no flow at all, but with anti-correlations come correlations and thus the opportunity for unrestricted flow of small and large materials via the emergent channels. If transport of materials through these emergent channels participates in the maintenance of the objects that constitute the borders, for example by supplying them with energy or clearing their waste products, then the alignment of patterns by the boundary constraints in turn becomes a (useful) constraint on those boundary constraints.

As an interesting example, the Voronoi-like tessellation of dark patches that gives the giraffe skin its distinctive patterning is geometrically related to an underlying vascular system. Particularly large arteries running between the patches supply a network of smaller arteries within the patches, which allow them to act as 'thermal windows' that efficiently radiate heat, and thus enable giraffes to thermoregulate in warm environments [30]. We note that giraffe panel substructures, not unlike subbarrel patterns in appearance, vary with the size of the panels, which in turn vary with the size of the animal in a manner predicted by reaction-diffusion modelling [8]. This raises the intriguing possibility that a relationship between the structure of

the vascular network and giraffe panel (and sub-panel) geometry may reflect a closure of constraints, co-opted for the thermal advantages it affords to these particularly large endotherms.

In terms of information processing, clustering of neurons to form tessellated patterns of cell density in and between brain nuclei constrains the transmission of signals between brain cells, and thus affords an opportunity for new information to be derived with reference to the underlying geometry, in turn enabling specific computations which facilitate survival [31–33]. The mammalian neocortex again provides a useful example. The arrangement of the barrels across the somatosensory cortex of rodents reflects the layout of the whiskers on their snouts, with cells of adjacent barrels responding most quickly and most strongly to deflection of adjacent whiskers. The relatively large size of the barrels, and the relatively slow velocities with which their efferents conduct action potentials, render downstream cells differentially sensitive to the relative timing of adjacent whisker deflections by virtue of their location with respect to barrel boundaries [34]. Neurons close to the borders respond selectively to coincident whisker deflections, and neurons that are closer to barrel A are selective for deflections of whisker B that precede deflections of whisker A by larger time intervals [35]. As such, the system can use the underlying geometry to compute the relative time interval between adjacent whisker deflections via place-coding [36, 37].

Within the additional cellular clusters that are formed via subbarrel patterning, neurons are tuned to a common direction of whisker movement [38], and somatotopically aligned maps of whisker movement direction subsequently emerge, such that deflection of whisker A towards B selectively activates neurons of barrel A that are closest to barrel B [39]. This particular alignment of information-processing maps is thought to occur by the specific constraints that the barrel and sub-barrel geometry imposes on the otherwise general-purpose processes of reaction-diffusion and Hebbian learning by which cortical maps self-organize [40, 41]. The relationship between these two patterns that is suggested by the present results provides a potential geometrical basis for the integration of sensory information. The alignment of within-barrel and between-barrel maps could render downstream cells sensitive to the coherence between single-whisker deflection directions and multi-whisker deflection intervals resulting from movement of tactile stimuli through the whisker field [42]. The net effect could be a representation of the 'tactile scene' that affords new possibilities for hunting and obstacle avoidance [43].

There are many other examples of tessellated patterns in the brain, including spots and stripes in primate primary visual areas, and barrel-like structures in the brainstem, thalamus, and extrasensory cortical areas in rodents, as well as in various cortical areas in moles, dolphins, manatees, platypus, monkeys, humans, and more (see [44] for an overview). The precise role that these patterned modular structures (fields, stripes, barrels, blobs) might play in cortical information processing is yet to be fully characterised [31, 32, 45]. However, in purely geometric terms, strong relationships between the shapes of cortical modules and the functional maps that they support have been well established. For example, iso-orientation contours radiating from the pinwheel centers that characterise topological maps of orientation preferences in primate primary visual cortex intersect with the boundaries of ocular dominance stripes at right angles [46, 47]. And numerous features of these functional maps have been successfully modelled in terms of reaction-diffusion dynamics (e.g., see [48–51]). Hence, considering only neocortical patterning, it seems the opportunities for constraint closure in the brain via computational geometry are abundant.

Montévil et al. [2] consider that reaction-diffusion dynamics introduce changes in the symmetries of a system that are *generic*, insofar as they derive from a restricted space of possibilities. By contrast, they consider organized biological wholes to be additionally defined by constraints that are *specific*, insofar as their dynamics depend on a history that spans

ontogenetic and phylogenetic timescales. In these terms, they suggest that modelling focused on deriving generic symmetries, which includes reaction-diffusion modelling, will ultimately fail to capture the individual accumulation of idiosyncrasies that characterize biological wholes (see also [52, 53]). While we agree with the importance of the history for understanding biological wholes, we do not agree that models formulated in terms of generic constraints are therefore fundamentally limited to describing only biological parts. The alignment of patterns between adjacent domains studied here constitutes a new (generic) symmetry that is invariant to the (specific) pattern that forms in either domain. Thus the local symmetry-breaking that generates patterns (i.e., Turing instability) also gives rise to symmetries that persist in the longer term (i.e., pattern alignment). As such, the opportunities that the alignment might afford to other processes (structural, transport, information-processing), persist at the same timescale at which the boundary conditions themselves persist. And if such processes can help to maintain the boundary conditions, for example by channeling an external supply to the cells that form the boundary, then the system achieves constraint closure, and hence the status of a biological whole (c.f. [1]). Indeed, the alignment between reaction-diffusion processes in tessellated domains, and the possibility for constraint closure that this affords, may prove to be a useful theoretical model through which to explore, by analysis *and* synthesis, the fundamentals of biological organization.

## Methods

### Numerical methods

Solutions to the reaction-diffusion equations (Eq 1) were obtained numerically on a discretized hexagonal lattice of grid points using the finite volume method described by [54], and a fourth-order Runge-Kutta solver was used to advance the solutions to a steady state (parameter values were chosen so that all solutions were eventually constant in time, i.e., patterns were stationary). Simulations were written in C++ with the help of the support library *morphologica* [55] (see also [14]). We verified that all simulation results reported were insensitive to the choice of spatial discretization (i.e., to the lattice density). A hexagon-to-hexagon distance of $O(10^{-3})$ on a domain whose spatial scale was normalized to be $O(10^0)$ was found to be sufficient. At this scale, domains contained $O(10^2)$ hexagonal grid-points and edge vectors with length $O(10^1)$.

No-flux boundary conditions were applied at the edges of the domains derived from a given tessellation, by setting all the normal components of the spatial gradient terms in Eq 1 to zero. Importantly, the *tangential* gradients at the boundary were not constrained, allowing the patterns on either side of the boundary to represent the pattern in the whole domain while patterns across adjacent boundary edges had no constraints that might be correlated. Solutions were considered to have converged when the mean of the absolute differences in field values sampled at intervals of 1000 timesteps fell below $\sim 10^{-10}$.

To compare the solutions along pairs of boundary vertices picked from two domains, the Pearson correlation coefficient was calculated. Vectors $\mathbf{x}_1$ and $\mathbf{x}_2$ each contained the solution values in the hexagonal grid points along (and one hexagon-to-hexagon distance inside) a boundary vertex from either domain. The boundary lines are shown in black in Figs 3–5, and the samples $\mathbf{x}_i$ were taken immediately adjacent to these boundary lines on either side. The field values of the solutions were always observed to vary smoothly along the edges and samples were thus not distorted by any boundary-related artefacts. These vectors were combined to compute the correlation coefficient, as $r(\mathbf{x}_1, \mathbf{x}_2) = \frac{\langle \mathbf{x}_1, \mathbf{x}_2 \rangle}{\|\mathbf{x}_1\|\|\mathbf{x}_2\|}$, where the numerator represents the Euclidean scalar product and the denominator gives the product of the Euclidean norms. This operation can be thought of as measuring the angle between two unit normal vectors. As

such, the distribution of the correlations may be considered a property of their distribution in a surrounding space—correlations lie on a hypersphere whose dimension is between 1 and $D - 1$, with $D$ the dimension of the containing space.

When comparing randomly matched edge vectors, the length of the shorter vector was increased to match that of the longer vector by linear interpolation. For all such results, the distances between interpolation points, even for the shorter vectors, were far smaller than the wavelengths that were observed to be amplified by pattern formation. Interpolating along the shorter vector was therefore appropriate, and preferable to downsampling along the longer vector, to avoid discarding information.

Code for running the simulations reported in this paper is available at https://github.com/ABRG-Models/Tessellations.

## Derivation of a test for the influence of boundary shape

Using $\alpha < 0.5$ as a threshold value for determining whether a set of patterns was constrained by the boundary shape can be justified analytically as follows. Consider a collection of edge patterns that are pinned to the vertices so that the maximal and minimal field values always appear at the two ends of each edge. The simplest (lowest mode) pattern that can be fitted to this constraint is a wave function $\cos(\theta)$, where $\theta$ ranges from $[0, \pi]$. The correlation $r$ between two such functions is given by their dot product

$$r = \pm \frac{\int_0^\pi \cos^2(\theta)d\theta}{\sqrt{(\int_0^\pi \cos^2(\theta)d\theta)}\sqrt{(\int_0^\pi \cos^2(\theta)d\theta)}}, \tag{3}$$

where the denominator gives the normalisation so that the vectors are of unit length, and hence Eq 3 returns either $r = 1$ or $r = -1$. If we relax the constraint that the patterns must be pinned to the vertices and allow the pattern along each edge to shift by $\phi \in [-\pi, \pi]$, where $\phi$ is drawn from a uniform distribution, then we need to evaluate

$$r = \frac{\int_0^\pi \cos(\theta)\cos(\theta + \phi)d\theta}{\sqrt{(\int_0^\pi \cos^2(\theta)d\theta)}\sqrt{(\int_0^\pi \cos^2(\theta + \phi)d\theta)}}. \tag{4}$$

By the simple change in variables, $\theta = \theta + \phi$, we can see that the denominator in Eq 4 is the same as in Eq 3. Expanding the numerator gives

$$\int_0^\pi \cos^2(\theta)\cos(\phi)d\theta - \int_0^\pi \cos(\theta)\sin(\theta)\sin(\phi)d\theta. \tag{5}$$

Either by explicitly evaluating the integral or by noting that $\cos(\theta)$ and $\sin(\theta)$ are anti-symmetric and symmetric about the midpoint of the range of integration, we can see that the second term vanishes, and hence that Eq 4 becomes

$$r = \frac{\cos(\phi)\int_0^\pi \cos^2(\theta)d\theta}{\int_0^\pi \cos^2(\theta)d\theta} = \cos(\phi). \tag{6}$$

Therefore the distribution of correlations consists of values given by $\cos(\phi)$, where $\phi$ is a random variable drawn from a uniform distribution over $[-\pi, \pi]$. This is exactly equivalent to the distribution of the dot product between two vectors on the unit circle with an angle between them that is chosen from a random uniform distribution. Thus it gives a symmetric beta distribution with $\alpha = 0.5$ (see e.g. [21] Ch. 4).

## Supporting information

**S1 Fig. Correlation of patterns in a system with the Turing mechanism.** We conducted a parameter sweep comparable to that presented in Fig 5, instead using the Schnackenberg reaction-diffusion system [22], whereby patterns form via the Turing mechanism. This system has no nonlinear diffusion term and its dynamics are driven by the magnitudes and ratios of two parameters that scale the linear diffusion operators, $D_A$ and $D_B$ (see [23]). **A** shows values of $\alpha$ obtained across a large portion of the parameter space, with the ratio of the two diffusion parameters, $D_B/D_A$, decreasing along the horizontal axis. The gradient in $\alpha$ corresponds well to that shown in Fig 5 using the Keller-Segel model. **B** shows examples of the field patterns for parameter values corresponding to the four corners in **A**.
(TIF)

## Acknowledgments

The authors thank Leah Krubitzer at the University California Davis for useful discussions, and Robert Schmidt at the University of Sheffield for useful discussion and comments on an earlier draft.

## Author Contributions

**Conceptualization:** John M. Brooke, Stuart P. Wilson.

**Formal analysis:** John M. Brooke, Sebastian S. James, Alejandro Jimenez-Rodriguez, Stuart P. Wilson.

**Funding acquisition:** Stuart P. Wilson.

**Investigation:** John M. Brooke, Sebastian S. James, Stuart P. Wilson.

**Methodology:** John M. Brooke, Stuart P. Wilson.

**Software:** John M. Brooke.

**Supervision:** Stuart P. Wilson.

**Validation:** John M. Brooke.

**Writing – original draft:** John M. Brooke, Stuart P. Wilson.

**Writing – review & editing:** John M. Brooke, Sebastian S. James, Stuart P. Wilson.

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
