## [Decision Letter · Decision Letter 0]

22 Sep 2021

Dear Dr. Wilson,

Thank you very much for submitting your manuscript "Biological action at a distance: Correlated pattern formation in adjacent tessellation domains without communication" for consideration at PLOS Computational Biology.

As with all papers reviewed by the journal, your manuscript was reviewed by members of the editorial board and by several independent reviewers. In light of the reviews (below this email), we would like to invite the resubmission of a significantly-revised version that takes into account the reviewers' comments.

We cannot make any decision about publication until we have seen the revised manuscript and your response to the reviewers' comments. Your revised manuscript is also likely to be sent to reviewers for further evaluation.

Sincerely,

Philip K Maini

Associate Editor

PLOS Computational Biology

Daniel Beard

Deputy Editor

PLOS Computational Biology

Reviewer's Responses to Questions

**Comments to the Authors:**

Reviewer #1: In this work, the authors are interested in understanding the interplay between boundary conditions and pattern formation on tessellated domains. They simulate reaction-diffusion equations on a range of tessellated domains, including collections of equilateral triangles, isosceles triangles, Voronoi cells, and Voronoi cells with rounded edges (assuming that pattern formation is independent in each cell). The authors propose a new means of elucidating whether or not domain boundaries actively constrained pattern formation by comparing the patterns that emerge across the tiles of the tessellation. They focus on the patterns with long wavelengths near the tile edges and show that the patterns sampled from adjacent tiles are correlated according to an overall bimodal distribution. It's an interesting, useful idea to compare the alignment of patterns in different regions, and the authors highlight how this could show up in biological applications. I have some questions about how the authors chose thresholds rigorously and how their results are related to patterns with small wavelengths. The Discussion is very broad; I recommend including more figures and more biological motivation, and reducing the Discussion breadth to better focus on the manuscript's contribution.

Main comments:

(1) I recommend moving the PDEs from Methods to the start of the Results section and discussing what the terms and notation in them means there.

(2) At the bottom of page 7 and beginning of page 8, I do not follow how the choice of the two alpha thresholds (alpha = 0.5 and alpha = 0.75) are made rigorously. Is this based on a single set of simulation conditions, for large wavelength patterns? I think it would be more meaningful if, for example, the authors repeated this study (constrained and control) across many more parameters, computed the resulting alignment distributions for all these cases, and then clustered the resulting distributions into two groups in an unsupervised way. If the resulting groups broke into constrained and control and held up the alpha thresholds, that would be stronger.

(3) In Figures 1 and 3, could the authors provide more intuition about how the reader should interpret the distribution for random edge pairs? Should I think of this as similar to just capturing alignment between patterns and boundaries? Could the authors test whether the distributions for adjacent edge pairs and random edge pairs are significantly different?

(4) The paper would benefit from more figures. The discussions around line 62 and line 70 in the Introduction could include figures (e.g., at line 70, an image of the rodent cortex, with barrels and sub-barrel patterns highlighted). I recommend including an illustration for the discussion on page 4 about type 1, 2, and 3 edges. In the discussion around line 145, I recommend referencing Figure 1.

(5) The language throughout is wordy (passive voice, long sentences, broad in the Discussion), and this can make it hard to understand the authors’ points. The breadth of the Discussion did not seem to match the results in the manuscript; my understanding of the main results of the paper are that the authors proposed a way of identifying the impact of boundaries on pattern formation in tessellated domains by comparing the patterns in adjacent tiles. The Discussion was much broader, and I would recommend reducing this. For example, the sentence “Thus the symmetry breaking by which pattern forms in the short term gives rise to symmetries that persist in the longer term” at line 437 did not seem to match the main results and contribution of the paper.

(6) The authors discuss several applications in the Discussion, and I would suggest moving some of these applications into the Introduction instead. This would help motivate the manuscript and provide concreteness. For example, at line 48 in the Introduction, when the authors discuss shape emerging subsequently to, simultaneously with, or independent of pattern formation, it would be useful to include some references and example biological applications for each of these options.

(7) At line 470, could the authors clarify what they mean by x_1 (for example, through a figure)? What does it mean that they are using the grid points along and just inside the boundary vertex? How far inside? At line 476, does the authors' choice to increase the longer vector by linear interpolation make sense when the pattern has a shorter wavelength (as in Fig. 4Bii)? It would be helpful if the authors could discuss this situation here.

Minor comments and typos:

(1) I recommend rewriting the sentence at line 30 (the last sentence in the Author Summary); it was unclear.

(2) I do not think the phrase “That is,…” at line 98 is a sentence.

(3) Around line 114, is there a reference for the discussion about where the reaction collects in the triangles? Additionally, related to one of my main comments, it would be clearer if the equations were introduced here rather than in Materials and Methods. Then the authors could discuss what “reaction collecting in the corners of the triangle” means in terms of the values of eta and c.

(4) I am not familiar with the arrow notation the authors use in the Results section (p_a ->).

(5) At line 149, should this be “with the bases’ perpendicular…”?

(6) At line 182, should it be “Fig 1D” rather than “(Fig 1D)”?

(7) At line 185, should it be “Dirichlet form” rather than “Dirichletform”?

(8) A t line 188, I think it should be “To establish whether” rather than “So to establish”.

(9) In the paragraph at line 200, it would be helpful if the authors could reference a figure showing the sub-barrel patterns and provide more biological background for readers who are not familiar with this application.

(10) At line 228, can the authors include how many images they analyzed?

(11) At line 260, the term beta(alpha, beta) is unclear. Could the authors choose a different variable name for the second beta?

(12) At line 290, there is a space missing at “Fig.2”.

(13) The sentence starting “By showing that” at line 330 in the Discussion is long, and I would recommend rewriting. As it is, it read to me like the main hypothesis was that patterns form by reaction-diffusion, but I think the main hypothesis is that domain boundaries actively constrain pattern formation by reaction-diffusion.

(14) At line 350, could the authors provide a reference and more biological background?

(15) At line 390, the sentence “The relationship between these two patterns…” is long and difficult to parse.

(16) At line 426, could the authors add the reference for the quote?

(17) At line 464, consistency-wise, the authors use “Eqs. 2”, but “Eq. 2” is used in all other places in the manuscript.

(18) In Materials and Methods, are r and C the same thing? Additionally, cosine is sometimes italicized and sometimes not throughout the equations on pg. 23.

(19) At line 496, could the authors rewrite this line (as is, it says that the distribution of correlations is cos(theta), but cos(theta) is not a distribution)?

(20) At line 497, I did not follow why “over [0,pi] or over [pi,0]” is needed; can this just be “over [0,pi]”?

(21) In reference [1], an accent character is missing in Montevil’s name.

Reviewer #2: In this paper, the authors continue their study of patterns in the whisker barrel system of the rodent. Their somewhat flashy title refers to the conjecture that if two domains have the same kinetics and diffusivities, then the patterns that appear via a Turing type instability will be similar and highly correlated along the edges even though there is no direct coupling. I think this is what they mean by "action at a distance." There are a few points I would like them to address. First -- there are similar phenomena in the temporal domain where two uncoupled limit cycles are subjected to correlated fluctuations and thus begin to synchronize. This is called the "Moran effect" and also "stochastic synchronization". I suggest that the present phenomena is closely related at least in a qualitative sense and they might want to discuss this literature. Secondly -- it is my impression that in large domains (or in the context of the paper, when the diffusivities and chemotaxis parameters are reduced, there can be many final patterns depending on the initial data. While they did address this in Fig 4, I remain skeptical that the correlations will be high when the pattern becomes more granular. Third--what if there are small differences in the parameters in each of the barrels? How sensitive if the correlation structure to such heterogeneities which i would imagine are pretty common in biological systems.

Other than these comments, I enjoyed this paper. They is an extra "are" in line 22.

**Have the authors made all data and (if applicable) computational code underlying the findings in their manuscript fully available?**

Reviewer #1: Yes

Reviewer #2: Yes

PLOS authors have the option to publish the peer review history of their article (what does this mean?). If published, this will include your full peer review and any attached files.

Reviewer #1: No

Reviewer #2: No
---

## [Decision Letter · Decision Letter 1]

16 Dec 2021

Dear Dr. Wilson,

Thank you very much for submitting your manuscript "Biological action at a distance: Correlated pattern formation in adjacent tessellation domains without communication" for consideration at PLOS Computational Biology.

As with all papers reviewed by the journal, your manuscript was reviewed by members of the editorial board and by several independent reviewers. In light of the reviews (below this email), we would like to invite the resubmission of a significantly-revised version that takes into account the reviewers' comments.

We cannot make any decision about publication until we have seen the revised manuscript and your response to the reviewers' comments. Your revised manuscript is also likely to be sent to reviewers for further evaluation.

Sincerely,

Philip K Maini

Associate Editor

PLOS Computational Biology

Daniel Beard

Deputy Editor

PLOS Computational Biology

Reviewer's Responses to Questions

**Comments to the Authors:**

Reviewer #1: Thank you to the authors for their careful revisions and thoughtful responses. I recommend acceptance.

Minor comments:

(1) Figure 1: Perhaps “Skin of giraffes” or “skin samples of giraffes” rather than “skins of giraffes”

(2) Figure 2A: I think the semicolon should be a comma: “patterns emerge, with extreme” rather than “patterns emerge; with extreme”.

Reviewer #2: The authors addressed all of my concerns, but i saw a comment in the paper on page 4 that had somehow escaped my earlier reading.

"most of the simulations reported we will set χ = 0," This makes me wonder how the authors got their numerical results. If chi=0, then in equation (1), n(x,t) -> 1 (assuming no-flux boundaries) since this is a then just a linear PDE and the solution can be exactly written down. In that case, f(n) -> constant and c(x,t)-> constant so that there will never be any pattern formation and thus there will be no patterns on the edge. If this statement is not true than i am happy to accept the paper, but if indeed chi=0, then the authors have not simulated the equations (1) since with chi=0, there can be no patterns

**Have the authors made all data and (if applicable) computational code underlying the findings in their manuscript fully available?**

Reviewer #1: None

Reviewer #2: Yes

PLOS authors have the option to publish the peer review history of their article (what does this mean?). If published, this will include your full peer review and any attached files.

Reviewer #1: No

Reviewer #2: No
---

## [Decision Letter · Decision Letter 2]

24 Feb 2022

Dear Dr. Wilson,

We are pleased to inform you that your manuscript 'Biological action at a distance: Correlated pattern formation in adjacent tessellation domains without communication' has been provisionally accepted for publication in PLOS Computational Biology.

Best regards,

Philip K Maini

Associate Editor

PLOS Computational Biology

Daniel Beard

Deputy Editor

PLOS Computational Biology

Reviewer's Responses to Questions

**Comments to the Authors:**

Reviewer #2: glad that this was helpful. The paper is more compelling

**Have the authors made all data and (if applicable) computational code underlying the findings in their manuscript fully available?**

Reviewer #2: None

PLOS authors have the option to publish the peer review history of their article (what does this mean?). If published, this will include your full peer review and any attached files.

Reviewer #2: **Yes: **Bard Ermentrout

---

## [Editor Report · Acceptance letter]

24 Mar 2022

PCOMPBIOL-D-21-01415R2 

Biological action at a distance: Correlated pattern formation in adjacent tessellation domains without communication

Dear Dr Wilson,

I am pleased to inform you that your manuscript has been formally accepted for publication in PLOS Computational Biology. Your manuscript is now with our production department and you will be notified of the publication date in due course.

With kind regards,

Livia Horvath
